# The Impact and Evolution of COVID-19 on Liver Transplant Recipients Throughout the Pandemic “Waves” in a Single Center

**DOI:** 10.3390/v17020273

**Published:** 2025-02-16

**Authors:** Clara Fernández Fernández, Blanca Otero Torrón, Mercedes Bernaldo de Quirós Fernández, Rafael San Juan Garrido, Cristina Martín-Arriscado Arroba, Iago Justo Alonso, Alberto Alejandro Marcacuzco Quinto, Óscar Caso Maestro, Félix Cambra Molero, Oana Anisa Nutu, Jorge Calvo Pulido, Alejandro Manrique Municio, Álvaro García-Sesma Pérez-Fuentes, Carmelo Loinaz Segurola

**Affiliations:** 1HBP and Transplant Surgery Unit, Department of General Surgery, Digestive Tract and Abdominal Organ Transplantation, Hospital Universitario “12 de Octubre”, 28041 Madrid, Spain; blanca.otero@salud.madrid.org (B.O.T.); mercedes.bernaldoquiros@salud.madrid.org (M.B.d.Q.F.); iagojusto@hotmail.com (I.J.A.); alejandro_mq@yahoo.es (A.A.M.Q.); oscarcasomaestro@hotmail.com (Ó.C.M.); felixcambra@gmail.com (F.C.M.); oanaanisa.nutu@salud.madrid.org (O.A.N.); jorge.calvo@salud.madrid.org (J.C.P.); alejandro.manrique@salud.madrid.org (A.M.M.); alvaro.garciasesma@salud.madrid.org (Á.G.-S.P.-F.); 2Unit of Infectious Diseases, Hospital Universitario “12 de Octubre”, 28041 Madrid, Spain; rafael.san@salud.madrid.org; 3Instituto de Investigación, Hospital “12 de Octubre” (imas12), 28041 Madrid, Spain; 4School of Medicine, Complutense University of Madrid, Avenida de Cordoba s/n, 28041 Madrid, Spain

**Keywords:** SARS-CoV-2, liver transplant recipients, liver transplantation, COVID-19 disease, COVID-19 waves, immunosuppression, epidemiology, vaccination

## Abstract

Liver transplant recipients (LTRs) have been considered a population group that is vulnerable to COVID-19 as they are chronically immunosuppressed patients with frequent comorbidities. This study describes the course of the SARS-CoV-2 disease from February 2020 to December 2023 along seven pandemic “waves”. We carried out an observational study on 307 COVID-19 cases in a cohort of LTRs with the aim of evaluating the changes in the disease characteristics over time and determining the risk factors for severe COVID-19. An older age and serum creatinine level ≥ 2 mg/dL were found to be risk factors for hospital admission and respiratory failure. The use of calcineurin inhibitors was a protective factor for death, hospitalization, and respiratory failure from COVID-19. One hundred percent of patients who died (N = 12) were on mycophenolate mofetil, which was a determinant for respiratory failure. Azathioprine was associated with admission to the intensive care unit (ICU) and with invasive mechanical ventilation (IMV). Vaccination was a protective factor for hospitalization, respiratory failure, and mortality. The severe COVID-19 rate was higher during the first five waves, with a peak of 57.14%, and the highest mortality rate (21.43%) occurred in the fourth wave. The IMV and ICU admission rates did not show significant differences across the periods studied.

## 1. Introduction

Since the outbreak of the coronavirus disease 2019 (COVID-19) pandemic caused by severe *acute respiratory syndrome coronavirus 2* (SARS-CoV-2), the number of cases registered by the WHO amounts to 776 million worldwide, with more than 7 million deaths [1]. Spain was the European country that suffered the fastest spread during the first months of the COVID-19 pandemic [2], which resulted in the temporary suspension of transplantation activity in multiple centers [3,4,5].

Solid organ transplant recipients have been considered a population group that is particularly vulnerable to COVID-19 as they are chronically immunosuppressed patients with frequent comorbidities [6,7,8,9]. The cumulative incidence of COVID-19 cases in liver transplant recipients (LTRs) has been approximately twice as high as in the non-transplanted population [7]. During the first months of the pandemic, the frequency of organ rejection and the overall in-hospital mortality increased in this group of patients [10].

The development and wide distribution of the different vaccines led to a change in the course of COVID-19. In LTRs, it has been shown to reduce the rates of infection, symptomatic disease [11], hospitalization [12], the need for admission to the ICU or mechanical ventilation [13], and mortality, especially in fully vaccinated patients [11,12,13].

The 12 de Octubre University Hospital is one of the 26 centers that carry out adult liver transplantation in Spain [14,15]. Since the beginning of the SARS-CoV2 pandemic, the consequences of COVID-19 on our solid organ transplant recipients have been studied [8,16,17,18,19].

The present study describes the course of the SARS-CoV-2 disease from February 2020 to December 2023, along successive pandemic periods or “waves”.

## 2. Patients and Methods

### 2.1. Aims and Objectives

This study aims to describe the characteristics of COVID-19-diagnosed LTRs throughout the successive pandemic stages or “waves”, evaluating the particularities and changes over the course of the disease and its management over time, and determining the risk factors for severe COVID-19 in LTRs and assessing those previously described in the literature.

### 2.2. Study Design

A single-center observational descriptive study was carried out on a cohort of 307 COVID-19 cases at the 12 de Octubre University Hospital in Madrid, from February 2020 to December 2023, in LTRs.

A case-registry database was created. The patients were recruited through communications to the transplant clinic, regardless of whether they required admission due to the infection, and patients who were admitted to the hospital due to COVID-19 were also recruited. Data were obtained by conducting a survey during a follow-up consultation in person or by telephone, and the information was bolstered using the hospital electronic medical record and medical reports from other centers.

### 2.3. Data Collection

Epidemiological data (sex, age), comorbidities (history of obesity, hypertension, chronic obstructive pulmonary disease, asthma, chronic and/or acute kidney disease, smoking), transplant-related data (indication and date of transplantation, immunosuppression), COVID-19-related data (symptoms, technique and date of diagnosis, treatment received, vaccination, association with altered liver enzymes), survival data (death due to COVID or other causes, overall survival), and follow-up data were collected.

The development of thrombotic events such as infarction, pulmonary thromboembolism, or deep vein thrombosis [20] in the context of COVID-19 was also investigated and/or questioned.

### 2.4. Definitions

Comorbidities: The following comorbidities were considered independently:
(1)Obesity (Body mass index (BMI) ≥ 30 kg/m^2^);(2)Acute or chronic kidney disease (presence at diagnosis of plasma creatinine ≥ 2 mg/dL due to acute kidney injury or as the baseline creatinine in patients with chronic kidney disease);(3)Chronic obstructive pulmonary disease (COPD);(4)Bronchial asthma;(5)Hypertension (HTN);(6)Type 2 diabetes mellitus (T2D);(7)Cardiovascular events or cardiovascular risk factors: arrhythmia, infarction, cardiomyopathy, heart failure or coronary artery disease, or the combination of HTN, T2D, and dyslipidemia;(8)Altered liver enzymes (elevation of liver enzymes (aminotransferase) above the upper limit of normal).

Diagnostic criteria for COVID-19: The following have been considered diagnostic for COVID-19:(1)A positive real-time polymerase chain reaction (RT-PCR) test for SARS-CoV-2;(2)A positive antigen detection test [21];(3)Patients with compatible symptoms and radiological findings suggestive of COVID-19 during the first weeks of the study, prior to the establishment of diagnostic tests [22];(4)A positive antibody test result before vaccination [23,24].

Severe COVID-19: Five scenarios were described [9]:(1)Required hospitalization;(2)Required admission to the intensive care unit (ICU);(3)Associated respiratory failure: baseline oxygen saturation ≤ 93%, partial blood pressure of oxygen (PaO_2_)/oxygen concentration (FiO_2_) ≤ 30 mmHg [25], or need for non-invasive ventilatory support;(4)Required invasive mechanical ventilation (IMV);(5)Died from COVID-19.

### 2.5. Study Time 

To assess the evolution of the characteristics of the disease throughout the pandemic, the patients have been grouped into seven periods or “waves”, taking as a reference the periods established by the Carlos III Institute [26] according to general incidence peaks. The Carlos III Institute is the main public organism of biomedical research, depending on the Ministry of Health in Spain.

−First period: from 1 March 2020, to 21 June 2020, the date on which the state of alarm in Spain ended alongside the first epidemic wave of COVID-19;−Second period: from 22 June to 6 December 2020;−Third period: from 7 December 2020, to 14 March 2021;−Fourth period: from 15 March 2021, to 19 June 2021;−Fifth period: from 20 June 2021, to 13 October 2021;−Sixth period: from 14 October 2021, to 27 March 2022;−Seventh period: from 28 March 2022, to 31 December 2023.

Before the 4th wave, COVID-19 variants were not studied routinely. The 4th period was characterized by Alpha variant circulation; the 5th period was dominated by the Delta variant, and the lasts waves were correlated with the Omicron variant [27].

### 2.6. Inclusion Criteria

−Patients with a history of liver transplantation for any indication, aged ≥ 18 years, who meet diagnostic criteria for COVID-19 were included.

### 2.7. Exclusion Criteria

−Patients with a history of transplantation of other solid organs or with multi-organ transplantation;−Patients < 18 years of age;−Patients for whom insufficient data were provided.

### 2.8. Statistital Analysis

The dataset of the study was summarized with descriptive statistical analysis. Quantitative variables were expressed as mean and standard deviation (SD) or median (p50) and interquartile range (IQR, p25–p75). The normality of the distributions was tested with Saphiro–Wilk test. Qualitative variables were expressed in absolute numbers (number of cases) and relative frequencies (percentage).

Relationships and comparison between groups ((1) patients hospitalized vs. COVID-19 outpatient management; (2) admitted or not admitted to the ICU; (3) with or without respiratory failure; (4) patients who required or did not require invasive mechanical ventilation; (5) patients who did or did not die from COVID-19; and (6) differences between the waves) were evaluated using the chi-square test (χ^2^) or Fisher’s exact test, Student’s *t*-test, Mann–Whitney U test, ANOVA, or Kruskal–Wallis test, as appropriate.

Binary logistic regression was used to analyze the risk factors associated with each one of the severe COVID-19 scenarios. The results were expressed as odds ratio (OR) with a 95% confidence interval (CI). Subsequent multivariate analysis was performed on those severe COVID-19 scenario groups with a N ≥ 30 (respiratory failure group and hospital admission group), considering risk factors that were found to be statistically significant in the univariate analysis and those considered relevant within the study objectives, using the stepwise regression method.

Survival analysis was performed using the Kaplan–Meier method. Log rank test was used to statistically compare the curves. Multivariable Cox regression analysis was used to explore predictors of death related to COVID-19.

The statistical software used was Stata for Windows version 16 (StataCorp. 2019. Stata Statistical Software: Release 16. StataCorp LLC, College Station, TX, USA). All analyses were conducted using an alpha significance level of 5% for a two-tailed *p*-value.

## 3. Results

### 3.1. Characteristics of the Study Population

A total of 542 patients who attended follow-up consultations or were admitted to the hospital during the study period were contacted in person and/or by telephone. Of these, 281 patients were identified as having been diagnosed with COVID-19. A total of 307 cases of COVID-19 were diagnosed, as 1 patient suffered two reinfections, and 24 patients had one reinfection. Totals of 24, 40, 21, 14, 10, 63, and 135 cases of disease were diagnosed in each period, respectively.

Of the 281 patients, 194 (69.04%) were male. The median age at diagnosis was 61 years (IQR 55–69). Regarding the patients’ medical histories, 13 (4.64%) had a history of COPD, 6 (2.14%) were asthmatic, 109 (38.93%) had hypertension, and 115 (41.07%) had T2D. Obesity was reported in 57 patients (20.73%), and 46 (16.85%) were smokers.

A total of 171 patients (61.29%) had a history of cardiovascular events or a combination of multiple cardiovascular risk factors. In 23 cases (7.52%), the creatinine at diagnosis was greater than 2 mg/dL.

The main indication for liver transplantation was alcoholic cirrhosis in 99 patients (35.23%), followed by HCV cirrhosis in 94 patients (33.45%) and hepatocellular carcinoma in 82 (29.18%).

### 3.2. Symptoms

The median time from transplantation to diagnosis was 94.46 months (IQR 41.44–185.21).

The symptoms at diagnosis are given in Table 1(A). The main symptoms were a cough in 120 cases (39.74%), asthenia in 98 case (32.45%), fever in 77 cases (25.50%), and myalgia in 73 cases (24.17%). In 60 cases (19.87%) the disease was asymptomatic.

In this study, there were two thrombotic events associated with COVID-19: one patient was diagnosed with a partial thrombosis of the inferior vena cava and another patient was diagnosed with an asymptomatic pulmonary thromboembolism. No cases of acute graft dysfunction associated with COVID-19 were identified.

### 3.3. Immunosupression

At the time of diagnosis, 189 patients (61.76%) were receiving calcineurin inhibitors (CNIs), 184 (60.13%) were receiving mycophenolate mofetil (MMF), 40 (13.07%) were receiving mTOR inhibitors (mTORi), 17 (5.56%) were on corticosteroid treatment, and 3 (0.98%) were receiving azathioprine (AZT).

### 3.4. Diagnosis of COVID-19

The diagnosis was established by antigen detection testing in 186 cases (60.59%), by RT-PCR in 110 cases (35.83%), by serological testing in 6 cases (1.95%), and by clinical diagnosis in 5 cases (1.63%). In asymptomatic patients, diagnosis was carried out by performing a PCR prior to a scheduled admission or test or by the patient obtaining a positive RT-PCR test after having been in close contact to confirmed COVID-19 cases.

A chest X-ray was performed in 84 cases, with a diagnosis of lobar pneumonia being obtained in 10 cases (3.64%) and bilateral pneumonia in 46 cases (16.73%). The radiological findings throughout the waves are given in Table 1(B).

### 3.5. Treatment for COVID-19

The treatment administered for COVID-19 is given in Table 2. A total of 64 cases (20.84%) required antibiotics. The administered antibiotics included beta-lactams, macrolides, quinolones, or oxazolidinones. Corticosteroids were administered in 45 cases (14.66%). The patient received prophylactic anticoagulation in 38 cases (12.38%). A total of 27 patients received Remdesivir 27 (8.79%), 13 patients received hydroxychloroquine (4.23%), 3 patients received Kaletra (0.98%), and 3 patients received Paxlovid (0.98%).

### 3.6. Vaccination

The median time from vaccination to the onset of symptoms was 5.87 months (IQR 3.23–9.03). In 211 cases (68.95%), the patient had received at least one dose of the vaccine at diagnosis, in 93 cases (30.39%) they had not received any dose. Table 3 shows the number of doses administered at diagnosis along the different periods.

### 3.7. Severe COVID-19

A total of 34 patients (11.07%) had respiratory failure, 5 (1.63%) required IMV, 72 (23.45%) were admitted to the hospital, and 12 of them (3.91%) were admitted to the ICU. Twelve patients (3.91%) died from COVID-19.

Characteristics of the patients who suffered respiratory failure, required mechanical ventilation, were hospitalized, were admitted to the ICU, and died from COVID-19 are given in Table 4, Table 5, Table 6, Table 7 and Table 8 respectively.

#### 3.7.1. Respiratory Failure

The patients with respiratory failure were significantly older, with a median age of 68 years (59–73), and had a higher median BMI, of 28 (IQR 25.5–29.7), an obesity rate of 50% (N = 6). HTN (67.65%, N = 23), and a more frequent history of cardiovascular events or risk factors (85.29%, N = 29) and D2T (67.65%, N = 23). The number of vaccinations was lower in this group, with a median of 0 doses (IQR 0–2). In 22 cases, the patient (64.70%) had not received any dose of the vaccine.

Thirty patients (88.24%) were on MMF and only 12 (35.29%) were on CNIs.

Univariate regression logistic analysis revealed that an older age (OR = 1.05, 95%CI: 1.02–1.09), the patient’s BMI (OR = 1.09, 95%CI 1.017–1.18), a creatinine level ≥ 2 mg/dL (OR = 4.12, 95%CI: 1.57–10.97), a history of cardiovascular events or risk factors (OR = 4.54, 95%CI: 1.7–12), T2D (OR = 3.48, 95%CI: 1.63–7.45), MMF (OR = 5.74, 95%CI: 1.97–16.76), bilateral pneumonia (OR = 112.5, 95%CI: 24.87–508.84), and lobar pneumonia (OR = 63, 95%CI 9.59–413.67) were risk factors for developing respiratory failure. Female sex (OR = 0.33, 95% CI: 0.12–0.89), CNIs (OR = 0.29, 95%CI: 0.14–0.62), and vaccination were protective factors (OR = 0.7, 95%CI: 0.54–0.92), as well as the number of months from vaccination to the onset of symptoms (OR = 0.7, 95%CI: 0.54–0.92).

#### 3.7.2. Invasive Mechanical Ventilation

One hundred percent of patients that needed IMV (N = 5) had a history of D2T and 20% (N = 1) were on AZT. Univariate regression logistic analysis showed that dyspnea (OR = 24.87, 95%CI: 2.71–228.14), AZT (OR = 37.35, 95%CI: 2.77–499.16), bilateral pneumonia (OR = 13.57, 95%CI: 1.37–133.71), and lobar pneumonia (OR = 37.25, 95%CI: 2.78–499.16) were risk factors for requiring IMV.

#### 3.7.3. Hospital Admission

The patients who required hospital admission were significantly older, with a median age of 64.5 years (IQR 56–71.5), and had a higher median BMI of 27.32 (IQR 24–29.7). The median length of hospital stay was 18 days (IQR 13–26). There were significantly more frequent comorbidities in this group, including kidney disease with creatinine ≥ 2 mg/dL (15.28%, N = 11), COPD (8.45%, N = 6), HTN (54.93%, N = 39), T2D (56.34%, N = 40), and a history of cardiovascular events or risk factors (76.06%, N = 54).

A total of 23.61% (N = 17) of the patients were on mTORi and 47.22% (N = 34) were on CNI.

The number of vaccinations was lower in this group, with a median of 0 doses (IQR 0–3). A total of 54.17% of the patients that were hospitalized (N = 39) had not received any dose of the vaccine. Univariate regression logistic analysis showed that vaccination was a protective factor for hospital admission (OR 0.23, 95%CI: 0.15–0.46).

The risk factors for hospital admission were age (OR = 1.04, 95%CI: 1.01–1.07), BMI (OR = 1.08, 95%CI: 1.02–1.15), creatinine ≥ 2 mg/dL (OR = 3.33, 95%CI 1.4–7.93), HTN (OR = 2.45, 95%CI 1.43–4.21), a history of cardiovascular events or risk factors (OR = 2.67, 95%CI: 1.46–4.89), and T2D (OR = 2.28, 95%CI: 1.33–3.9). mTORi and MMF were risks factors for admission (OR = 2.84, 95%CI: 1.42–5.67 and OR = 2.01, 95%CI 1.13–3.58, respectively) while CNI was a protective factor (OR = 0.46, 95%CI: 0.27–0.78).

#### 3.7.4. Admission to the ICU

The five patients admitted to the ICU were male and one of them (20%) was in AZT, which resulted in a risk factor of admission to the ICU in the univariable regression logistic analysis (OR = 13.27, 95%CI: 1.12–157.67). Dyspnea and fever were the only symptoms related to ICU admission (OR = 21.17, 95%CI: 5.47–81.85 and OR = 4.4, 95%CI 1.35–14.3), and radiological findings of bilateral pneumonia (OR = 28.5, 95%CI 3.34–243.27) and lobar pneumonia (OR = 21.11, 95%CI 1.22–365.44) were also risk factors.

#### 3.7.5. Multivariate Analysis of Respiratory Failure and Hospital Admission

Multiple models were evaluated to carry out a multivariate analysis considering risk factors described in the univariate analysis. The results are shown in Table 9.

For respiratory failure (N = 34), the model with the largest area under the ROC curve (0.851) included age (OR = 1.06, 95%CI: 1–1.1), HTN (OR = 3.69, 95%CI: 1.61–8.45), creatinine ≥ 2 mg/dL (OR = 5.54, 95%CI: 1.62–17.52), immunosuppression with MMF (OR = 2.73, 95%CI: 1.93–14.71), and vaccination (OR = 0.16, 95%CI 0.072–0.37).

For hospital admission (n = 72), the model with the largest area under the ROC curve (0.896) included creatinine ≥ 2 mg/dL (OR = 4.29, 95%CI 1.35–13.58), HTN (OR = 3.25, 95%CI: 1.54–6.89), dyspnea (OR = 18.83, 95%CI: 7.61–46.56), fever (OR = 2.8, 95%CI: 1.9–8.74), immunosuppression with mTOR inhibitors (OR = 2.8, 95%CI: 1.07–7.32), and vaccination (OR 0.2, 95%CI 0.09–0.44).

### 3.8. COVID-19 Mortality

Twelve patients died due to COVID-19 infection. Characteristics and a univariant survival analysis of these patients are given in Table 10. These patients were significantly older, with a median age of 68 years (IQR 61.5–71.5). The frequencies of these patients having a history of COPD, HTN, cardiovascular events or risk factors, T2D, and kidney injury were significantly higher, with rates of 16.67% (N = 2), 66.67% (N = 8), 91.67% (N = 11), 83.33% (N = 10), and 33.33% (N = 4), respectively.

The number of doses of the vaccine at diagnosis was lower in the patients that died from COVID-19, with a median of 0 (IQR 0–1.5).

One hundred percent of the patients who died due to COVID-19 were on MMF, and only 16.67% (N = 2) were on CNIs. The variables related to mortality from COVID-19 in the univariate survival analysis were obesity (HR = 3.94, 95%CI: 1.27–12.22), creatinine ≥ 2 mg/dL (HR = 6.9, 95%CI: 2.08–23.02), COPD (HR = 4.64, 95%CI: 1.02–21.22), and T2D (HR = 7.39, 95%CI: 1.62–33.76). CNIs (HR = 0.12, 95%CI: 0.026–0.55) and vaccination (HR = 0.56, 95%CI 0.32–0.83) were protective factors.

### 3.9. Evolution of Severe Disease Throughout the COVID-19 Waves

Differences in the incidence of severe COVID-19 scenarios along the waves were analyzed. The results are shown in Table 10. Globally, the severe COVID-19 rate was higher during the first five waves, with a peak of 57.14% (N = 8) in the fourth period and a great decrease in the sixth and seventh periods. Vaccination programs began in the third wave, with a subsequent progressive increase in the number of vaccinated patients and the doses administered.

The number of vaccinated patients, diagnosed cases by date, and deaths from COVID-19 are illustrated in Figure 1. In the sixth and seventh waves, we observed a significant reduction in the rates of respiratory failure, 6.35% (N = 4) and 1.48% (N = 2), respectively, and in the rates of hospitalization, 17.46% (N = 11) and 9.63% (N = 13), respectively.

This contrasts with the incidence observed in the fourth and fifth waves, which reached 42.86% (N = 6) of patients with respiratory failure and 64.29% (N = 9) of hospitalized patients, similarly to the mortality, which reached the highest rate in the fourth wave, 21.43% (N = 3). Kaplan–Meier curves for the survival across waves are illustrated in Figure 2.

Surprisingly, the VMI and ICU admission rates did not show significant differences across the periods, although they remained below 15% during the entire pandemic.

## 4. Discussion

Repercussions of the pandemic on liver transplant activity: The COVID-19 pandemic has repercussed on liver transplantation activity, with a decrease in the number of organ donors and even the temporary suspension of the activity in several centers [3,4]. A study which analyzed the database from the Global Observatory for Organ Donation and Transplantation between 2019 and 2020 observed a global decrease of 17.5% in the number of donors, with an 11.3% decrease in liver transplants [28]. In an observational study which analyzed the transplant activity across 22 countries during the same period, a global decrease of 16% was observed [5]; this decrease was more notable during the first months of the pandemic.

Inayat F et al. [10] studied, retrospectively, a cohort of over 15,000 patients who received a liver transplant that were hospitalized between 2019 and 2020. They found that the COVID-19 pandemic did not translate into an increase in the risk of hospitalization in these patients. It was however shown to have an effect of increasing the intrahospital mortality and organ rejection rates among the patients hospitalized during said period.

In Spain, a study was conducted on organ donation among patients with a positive CRP for SARS-CoV-2 in 69 organ recipients who had negative CRPs. The CRP resulted positive in four of the recipients (5.8%) within 20 days of transplantation, without it being able to be attributed to the transplant. None among the 18 recipients of a liver transplant suffered major complications that could be linked to infection in the donor [29]. Another multicenter study evaluated viral RNA in the liver biopsy of 10 grafts, with negative results being obtained in all of them [30].

Characteristics of the transplanted patients: Kulkarni et al. conducted, in 2021, the only meta-analysis published to date [1], including 18 observational studies with a total of 1522 patients. According to this study, the most common etiologies for transplant were viral (38%), alcohol (22.23%), NASH (2.5%), autoimmune (7.4%), and hepatocarcinoma (5.26%). In our study, viral liver disease, alcoholic liver disease, and hepatocellular carcinoma were the primary reasons for transplantation. The comorbidities that were most frequently registered were hypertension (44.3%), diabetes (39.4%), cardiovascular disease (16.43%), renal failure (13.7%), pulmonary disease (9%), and obesity (23.6%) [9]. In our study, the frequencies of diabetes mellitus, hypertension, and obesity were similar, with frequencies of 41.07%, 38.93%, and 20.73%, respectively. The frequency of pulmonary disease (COPD or bronchial asthma) was 6.78%, while the frequency of cardiovascular events or a combination of multiple cardiovascular risk factors was up to 61.29%.

COVID-19 in orthotopic liver transplant: It was during the first wave of the disease that a large series of cases of adult liver transplant patients infected with COVID-19 were published, contributing greatly to the current knowledge of the particularities of the disease in this subgroup of patients [8,19,20,31,32,33].

The most common symptoms described in LT are fever (49.7%), cough (43.79%), and dyspnea (29.7%), with a 27.26% incidence rate of gastrointestinal symptoms [9], myalgia or asthenia (36.66%), anosmia (36.66%), or dysgeusia (33.33%) [7].

In our cohort, the predominant symptoms were not severe, with cough, asthenia, and myalgias encompassing the majority. Gastrointestinal symptoms (nausea, vomiting, and diarrhea) were only alluded to by 14.9% of patients. Among our patients, 25.5% referred to fever and 14.9% referred to dyspnea, with an observed progressive decrease in this latter symptom in the subsequent waves. Presenting at the time of diagnosis with asthenia, dyspnea, fever, or anorexia has proven to be a risk factor for developing respiratory insufficiency in this study, with dyspnea being the only symptom that proved to be a risk factor for the need of mechanical ventilation. The presence of fever and dyspnea were the only symptoms at the time of diagnosis that resulted in a higher risk of requiring intensive care.

The median time between receiving the transplant and COVID-19 diagnosis is variable, from 5.7 to 13.1 years [7,9,34]. Guarino M et al. observed that it was significantly shorter in patients with asymptomatic COVID-19 [2]. These, however, did not show any differences in regard to comorbidities or immunosuppressive treatment. In a 16-patient series, Eren-Kutsoylu et al. [34] found a correlation between respiratory symptoms or fever and the need for hospitalization. In this series, 14 patients remained asymptomatic (46.67%), in contrast with the series by Colmenero et al., which included 7/111 patients who were asymptomatic (6.3%) [8]. In this study, we found 19.87% of patients to be asymptomatic, with a variable frequency among different waves that was higher in the third and sixth (25.81–33.33%) versus the first and second (16.67–19.87%), all of which had a higher frequency than the last one (14.50%).

Kulkarni et al. describe a 72% rate of admissions, higher than in non-transplanted patients and three times higher than the one observed in our study, and a cumulative incidence of intensive care unit admission of 16%, compared to the 3.9% in our study [9].

A mean hospital stay for LTRs of between 8 and 11 days was reported [7,32,35], similar to that of the non-transplanted controls [36] and shorter than that found in our study, which was 18 days. The incidence of invasive mechanical ventilation described in the above-mentioned meta-analysis was 21.1%, again higher in LTRs than in the non-transplanted controls [9]. In our study, only five patients (1.63%) required intubation.

The lower rate of severe COVID-19 in this study is justified by the inclusion in the same of a higher number of patients towards the end of the pandemic. As can be observed in Table 10, the rate of severe COVID-19 decreases towards the sixth and seventh waves, at which time over 60% of the cases were diagnosed. This draws attention, however, to the fact that the rate of intensive care unit admissions and the need for mechanical ventilation have shown no statistically significant differences during these periods.

In regard to thrombotic complications, the incidence among transplanted patients lies around 6% [9]. Mansoor et al. [20] found no differences when comparing this incidence rate to the thrombotic rate of the general public.

The median age at the time of diagnosis is lower (59.55 years) in transplanted patients compared to their controls (63.2 years). The median age at the time of diagnosis was similar in our study, 61 years. Colmenero et al. and Belli et al. describe that patients with less severe symptoms from COVID-19 were younger, had fewer comorbidities, and more frequently received immunosuppressive therapy with tacrolimus [8,19]. When compared to the general public, it can be seen that the course of the disease has not shown more severity in LTRs [7,33].

In this cohort, the median age and BMI were significantly higher in patients who suffered respiratory insufficiency or who required a longer hospital stay. In the logistic regression analysis, older patients and those who had a high BMI, renal failure with creatinine over 2 mg/dL, a history of cardiovascular events or risk factors, HTN, and T2D have been shown to be risk factors for respiratory insufficiency or hospital admission, while female sex proved to be a protective factor.

A history of COPD was a risk factor for hospital admission, as well as mTORi immunosuppressive therapy, while the number of vaccinations before diagnosis proved to be a protective factor for hospital admission and the development of respiratory insufficiency.

The time between vaccination and the start of symptoms was significantly shorter in patients who later developed respiratory insufficiency, but no significant differences were observed for hospital admission, admission in the ICU, or need for intubation.

Altered x-rays were observed to be a significant risk factor for severe COVID-19 (hospital admission, intensive care unit admission, respiratory insufficiency, or need for mechanical ventilation).

COVID-19-related mortality in orthotopic liver transplant: The factors related to death from COVID-19 have been exhaustively studied. The results for mortality in LT are heterogenous in the published series [9], with a 16.5% incidence and similar values when compared to non-transplanted patients. Webb et al., in a multicenter study with 151 LTs among 18 countries, described a significantly lower mortality rate in LT (19% versus 27% in non-LT; *p* = 0.046). The main causes of death that are described are COVID-19-related complications (62.54%) and respiratory failure (29.88%) [8,9,19].

Increased levels of transaminases during COVID-19 infection have been documented [9]. Rabiee et al. described an association between changes in hepatic biochemistry and increase in mortality in a series of 112 LTRs [33]. Despite this, no differences were apparent in this study when analyzing alterations in hepatic biochemistry and death by COVID-19.

Concerning acute renal failure, a 33.22% incidence rate was reported in LTRs with COVID-19 [9], indicating that the increase in basal serum creatinine concentration to be a risk factor for severe COVID-19 and mortality [31,36].

Other contributing factors to COVID-19 mortality are the presence of malignant extrahepatic neoplasia [32], advanced age [8,19,31], dyspnea at the time of diagnosis [8], male sex, D-dimer or ferritin level elevation, and lymphopenia [8,32,37].

In our study, the variables related to death by COVID-19 were obesity, renal failure, COPD, and DM2, while vaccination and immunosuppressive therapy with CNIs were protective factors. All the patients admitted to the ICU and those who required IMV were male as well as 10/12 (83.33%) of the patients who died from COVID-19.

Immunosuppression: The immunosuppression schemes in previously published studies are based on CNIs, cyclosporine, MMF, or mTORi in monotherapy or in combination [8,9,19,31,32]. Colmenero et al. showed the prejudicial and dose-dependent effect of MMF immunosuppression [8], possibly in relation to a synergic mechanism in cytotoxicity, on CD8+ with the COVID-19 virus. They did not find worse results in terms of severity in immunosuppressed patients with CNIs or mTORi. Belli et al. did prove an association between CNIs and reduced mortality which could be related to an inhibitory mechanism of viral replication [19]. mTOR inhibitors, AZT, and MMF can induce cytopenia, which could worsen the course of infection since lymphopenia is a parameter associated with severe COVID-19 [38,39].

A change in immunosuppressive therapy has not been shown to have an impact on mortality [19,31,34,37,40].

The consensus among the AASLD and the EASL is that an individualized evaluation of each case of COVID-19 infection, preferably reducing the dosage of mycophenolate in patients at a higher risk of developing severe forms of the disease, is ideal [41,42].

Vaccination against COVID-19: LTRs have shown a lower prevalence of anti-SARS-CoV-2 antibodies in the long term when compared to the general public, with higher levels of antibodies being seen in vaccinated patients [43,44,45] and in patients with a longer period between receiving the transplant and being infected by COVID-19 [43].

Currently, 12 vaccinations comprise the Emergency Use Listing of the WHO, including inactivated viral vaccines and protein-based and DNA or RNA viral vector-based vaccines [46,47]. The development and prompt distribution of the different vaccines has meant a change in the course of COVID-19, protecting against more severe forms and death. In LTRs, it has been proven to reduce the rates of infection, symptomatic disease [11], hospitalization [12], the need for intensive care and mechanical ventilation [13], and mortality, particularly in patients with a complete course [11,12,13]. The associations for the study of liver disease recommend a complete vaccination, as well as a booster dose to achieve or maintain immunity in LTRs [48,49]. This subgroup exhibits a lower immunological response than the general public due to receiving immunosuppressive therapy—particularly with MMF-—achieving a higher seroconversion dose in patients who received repeated doses [44,50,51,52]. Long-term LTRs show a better immunological response than recent recipients [53].

This study shows a lower prevalence of severe COVID-19 in vaccinated patients, with a higher number of doses in those patients who did not present with complications.

Treatment for COVID-19 infection in LTRs: Together with general measures such as hand washing, the use of facemasks, and social distancing, multiple drugs have been evaluated as potential treatments for COVID-19, including hydroxychloroquine, antibiotics, antivirals, steroids, immunomodulators, anticoagulants, and plasma [9].

Antimalarials such as hydroxychloroquine were considered during the first stages of the pandemic for their immunomodulator and antiviral effects, but their efficacy has not been clearly proven [17,54]. Remdervir, a SARS-CoV-2 RNA polimerase inhibitor, has been widely used since it is a medication with an acceptable safety level, but it has not shown strong benefits in patients with a solid organ transplant [55,56], which induced the WHO to discontinue its use in hospitalized patients [55]. Nonetheless, it is still recommended for outpatient treatment in those with a high risk of hospitalization [57]. Favipiravir [58] and Lopinavir/Ritonavir, which showed in vitro activity against SARS-CoV-2 [54,59], are discouraged. Dexamethasone has proved to reduce mortality in those who require mechanical ventilation [60]. Tocilizumab, an IL-6 receptor inhibitor, is considered a safe medication in transplanted patients, even though it must be used carefully owing to its hepatotoxicity [54,59]. Other monoclonal antibodies such as bamlanivimab, casirivimab, and imdevimab have proven to reduce the viral load in patients with non-severe COVID-19 [55].

In our study, we observed that antimalarials, interferon, and kaletra were used during the first period in selected cases. When bacterial superinfection was suspected, empiric antibiotics were prescribed and later switched according to antibiogram. There was no immunosuppressive adjustment depending on the antibiotic administration, but all the patients had repeated immunosuppression level controls to maintain appropriate dosing during their hospitalization. Prophylactic anticoagulation was prescribed in 50% of the patients up to the fifth wave, with a decrease in the last two waves, following the same tendency as corticoid treatment.

Tocilizumab and remdesivir were used in 2.61% (N = 8) and 8.79% (N = 27) of the patients, respectively. The use of Paxlovid requires careful monitoring in immunosuppressive therapy; therefore, its use is limited in transplant patients [61]. In our study, it has only been used in 1.02% of patients (N = 3).

Study limitations: This study has limitations due to its observational and retrospective design. Data were missing, as their report in the electronic medical records was not routinely and prospectively collected for research, which led to the exclusion of potential study subjects. Recall bias may have occurred, as participants may inaccurately remember their experiences, as they were surveyed after the infection, with the exception of those admitted to the hospital. Despite collecting data from a considerable number of patients, this study evaluates a cohort in a single center, which limits its external validity and implies a selection bias.

## Figures and Tables

**Figure 1 viruses-17-00273-f001:**
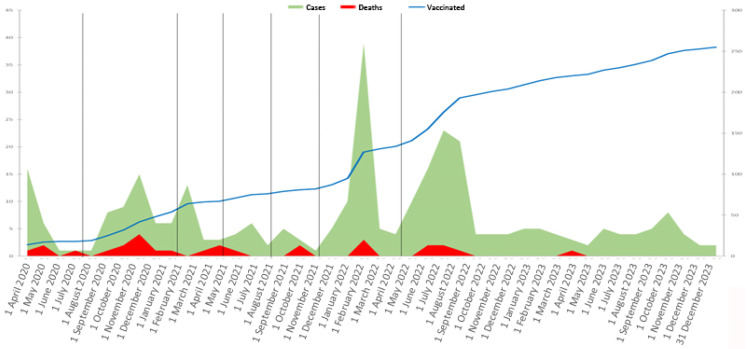
Vaccinated patients, diagnosed cases by date, and deaths from COVID-19.

**Figure 2 viruses-17-00273-f002:**
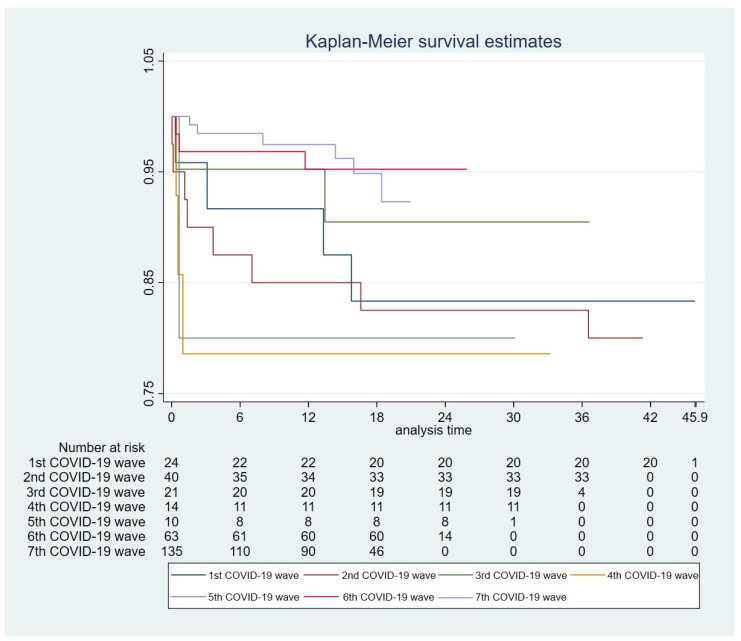
Kaplan–Meier curves for survival across COVID-19 waves.

**Table 1 viruses-17-00273-t001:** Symptoms at diagnosis and chest X-ray findings.

A. Symptoms at Diagnosis									*p*-Value
Period		1st Wave	2nd Wave	3rd Wave	4th Wave	5th Wave	6th Wave	7th Wave	
**Number of cases**	N = 302 (5 lost cases)	24	40	21	14	10	63	135	
**Odynophagia**	30 (9.93%)	1 (4.17%)	0	1 (4.76%)	1 (7.14%)	1 (10%)	5 (8.06%)	21 (16.03%)	0.068
**Rhinorrhoea**	46 (15.23%)	0	1 (2.5%)	2 (9.52%)	0	1 (10%)	14 (22.58%)	28 (21.37%)	0.004
**Nausea or vomiting**	13 (4.30%)	1 (4.17%)	4 (10%)	2 (9.52%)	0	0	2 (3.23%)	4 (3.05%)	0.39
**Asthenia**	98 (32.45%)	5 (20.83%)	8 (20%)	5 (23.81%)	4 (28.57%)	4 (40%)	14 (22.58%)	58 (44.27%)	0.011
**Myalgias**	73 (24.17%)	2 (8.33%)	12 (30%)	4 (19.05%)	2 (14.29%)	2 (20%)	11 (17.74%)	40 (30.53%)	0.15
**Dyspnea**	45 (14.90%)	7 (29.17%)	10 (25%)	2 (9.52%)	3 (21.43%)	1 (10%)	11 (17.74%)	11 (8.40%)	0.043
**Ageusia**	11 (3.64%)	0	2 (5%)	0	0	3 (30%)	3 (4.84%)	3 (2.29%)	<0.001
**Anosmia**	15 (4.97%)	0	4 (10%)	1 (4.76%)	0	4 (40%)	2 (3.23%)	4 (3.05%)	<0.001
**Chest pain**	12 (3.97%)	2 (8.33%)	1 (2.5%)	1 (4.76%)	1 (7.14%)	0	4 (6.45%)	3 (2.29%)	0.65
**Diarrhoea**	32 (10.60%)	7 (29.17%)	7 (17.5%)	2 (9.52%)	1 (7.14%)	1 (10%)	4 (6.45%)	10 (7.63%)	0.039
**Febricle**	58 (19.21%)	2 (8.33%)	0	2 (9.52%)	1 (7.14%)	3 (30%)	13 (20.97%)	37 (28.24%)	0.001
**Fever**	77 (25.50%)	6 (25.00%)	16 (40%)	7 (33.33%)	1 (7.14%)	2 (20%)	15 (24.19%)	30 (22.9%)	0.22
**Anorexia**	14 (4.64%)	0	3 (7.5%)	2 (9.52%)	0	0	3 (4.84%)	6 (4.58%)	0.65
**Headache**	33 (10.93%)	1 (4.17%)	1 (2.5%)	2 (9.52%)	1 (7.14%)	0	10 (16.13%)	18 (13.74%)	0.22
**Conjunctivitis**	1 (0.33%)	0	1 (2.5%)	0	0	0	0	0	0.36
**Expectoration**	26 (8.61%)	1 (4.17%)	1 (2.5%)	1 (4.76%)	2 (14.29%)	0	6 (9.68%)	15 (11.45%)	0.44
**Cough**	120 (39.74%)	14 (58.33%)	16 (40%)	4 (19.05%)	2 (14.29%)	3 (30%)	18 (29.03%)	63 (48.09%)	0.006
**Asymptomatic**	60 (19.87%)	4 (16.67%)	7 (17.50%)	7 (33.33%)	4 (28.57%)	3 (30%)	16 (25.81%)	19 (14.50%)	0.26

**B. Chest X-ray findings**									*p*-value
**Period**		**1st wave**	**2nd wave**	**3rd wave**	**4th wave**	**5th wave**	**6th wave**	**7th wave**	<0.001
**Number of cases**	N = 275 (32 lost cases)	24	40	21	14	10	63	135	
**Not performed**	191 (69.45%)	7 (30.43%)	24 (61.54%)	11 (57.89%)	4 (30.77%)	4 (57.14%)	42 (77.78%)	99 (82.5%)	
**Normal**	28 (10.18%)	5 (21.74%)	0	1 (5.26%)	2 (15.38%)	0	4 (7.41%)	16 (13.33%)	
**Bilateral pneumonia**	46 (16.73%)	9 (39.13%)	13 (33.33%)	7 (36.84%)	6 (46.15%)	3 (42.86%)	6 (11.11%)	2 (1.67%)	
**Lobar pneumonia**	10 (3.64%)	2 (8.70%)	2 (5.13%)	0	1 (7.69%)	0	2 (3.70%)	3 (2.50%)	

**Table 2 viruses-17-00273-t002:** Drugs administered in each period.

Period		1st Wave	2nd Wave	3rd Wave	4th Wave	5th Wave	6th Wave	7th Wave
**Number of cases**	N = 307	24	40	21	14	10	63	135
**Antibiotics**	50 (16.29%)	11 (45.83%)	13 (32.50%)	2 (9.52%)	5 (35.71%)	2 (20.00%)	7 (11.11%)	10 (7.41%)
**Anticoagulation**	38 (12.38%)	5 (20.83%)	12 (30.00%)	4 (19.05%)	6 (42.86%)	5 (50.00%)	5 (7.94%)	1 (0.74%)
**Antimalarials**	13 (4.23%)	12 (50.00%)	1 (2.50%)	0	0	0	0	0
**Corticosteroids**	45 (14.66%)	3 (12.50%)	13 (32.50%)	4 (19.05%)	5 (35.71%)	5 (50.00%)	10 (15.87%)	5 (3.70%)
**Interferon**	2 (0.65%)	2 (8.33%)	0	0	0	0	0	0
**Kaletra**	3 (0.98%)	3 (12.50%)	0	0	0	0	0	0
**Tocilizumab**	8 (2.61%)	2 (8.33%)	0	1 (4.76%)	3 (21.43%)	1 (10.00%)	1 (1.59%)	0
**Remdesivir**	27 (8.79%)	0	3 (7.50%)	0	1 (7.14%)	1 (10.00%)	4 (6.35%)	18 (13.33%)
**Paxlovid**	3 (1.02%)	0	0	0	0	0	0	3 (2.46%)

**Table 3 viruses-17-00273-t003:** Vaccination throughout COVID-19 waves.

Period		1st Wave	2nd Wave	3rd Wave	4th Wave	5th Wave	6th Wave	7th Wave
**Number of cases**		24	40	21	14	10	63	135
**Number of vaccines at diagnosis**								
0	93 (30.39%)	24 (100%)	40 (100%)	21 (100%)	8 (57.14%)	0	0	0
1	16 (5.23%)	0	0	0	5 (35.71%)	1 (10%)	5 (8.06%)	5 (3.70%)
2	38 (12.42%)	0	0	0	1 (7.14%)	9 (90%)	17 (27.42%)	11 (8.15%)
3	92 (30.07%)	0	0	0	0	0	38 (61.29%)	54 (40%)
4	54 (17.65%)	0	0	0	0	0	2 (3.23%)	52 (38.52%)
5	13 (4.25%)	0	0	0	0	0	0	13 (9.63%)
**Number of vaccines at diagnosis: median (Interquantile range (IQR))**	3 (0–3)	0	0	0	0 (0–1)	2 (2–2)	3 (2–3)	3 (3–4)
**Time from vaccine to diagnosis (months)**	5.87 (3.23–9.03)				0.49 (0.3–0.52)	2.89 (1.77–3.28)	3.61 (3.11–4.98)	7.34 (4.23–10.62)

**Table 4 viruses-17-00273-t004:** Respiratory failure in LTRs diagnosed with COVID-19.

					Logistic Regression Analysis	
		Respiratory Failure				
	Total (N = 307)	No (N = 273)	Yes (N = 34)	*p*-Value	Odds Ratio (OR) (*p*-Value)	95% Confidence Interval (CI)
**Age (years), median (IQR)**	61 (54–68)	61 (53–68)	68 (59–73)	0.002	1.05 (0.002)	1.02–1.09
**Body mass Index (BMI) (kg/m^2^), median (IQR)**	26.3 (23–29)	26.05 (22.7–29)	28 (25.5–29.7)	0.026	OR 1.09 (0.017)	1.017–1.18
**Male sex, N (%)**	209 (68.08%)	180 (65.93%)	29 (85.29%)	0.022		
**Female sex, N (%)**	98 (31.92%)	93 (34.07%)	5 (14.71%)		0.33 (0.028)	0.12–0.89
**Number of vaccines at diagnosis**	3 (0–3)	3 (0–3)	0 (0–2)	<0.001	0.52 (<0.001)	0.39–0.69
**Vaccination N (%)**	211 (68.95%)	199 (73.16%)	12 (35.29%)	<0.001	0.2 (<0.001)	0.094–0.42
**Time (months) from vaccination to the onset of symptoms, median (IQR)**	5.87 (3.23–9.03)	6.46 (3.31–9.11)	2.89 (0.62–4.36)	0.003	0.7 (0.011)	0.54–0.92
**Time (months) from transplantation to the onset of symptoms, median (IQR)**	94.46 (41.44–185.21)	94.46 (41.44–184.49)	95.08 (51.28–188.75)	0.79		
**Kidney failure (Creatinine ≥ 2 mg/dL)**	23 (7.52%)	16 (5.88%)	7 (20.59%)	0.002	4.12 (0.004)	1.57–10.97
**Chronic Obstructive Pulmonary disease (COPD), N (%)**	13 (4.25%)	10 (3.68%)	3 (8.82%)	0.16		
**Asthma, N (%)**	6 (1.96%)	6 (2.21%)	0	0.38		
**Hypertension (HTN), N (%)**	117 (38.24%)	94 (34.56%)	23 (67.65%)	<0.001	3.96 (<0.001)	1.85–8.47
**Cardiovascular events and/or risk factors**	181 (59.34%)	152 (56.09%)	29 (85.29%)	0.001	4.54 (0.002)	1.7–12
**Type 2 Diabetes mellitus (T2D), N (%)**	125 (40.85%)	102 (37.50%)	23 (67.65%)	<0.001	3.48 (<0.001)	1.63–7.45
**Altered liver enzymes, N (%)**	20 (6.54%)	16 (6.81%)	4 (5.63%)	0.73		
**Smokers, N (%)**	50 (16.72%)	44 (16.54%)	6 (18.18%)	0.53		
**Asthenia, N (%)**	98 (32.45%)	81 (30.22%)	17 (50.00%)	0.02	2.31 (0.023)	1.12–4.75
**Dyspnea, N (%)**	45 (14.9 %)	25 (9.33%)	20 (58.82%)	<0.001	13.88 (<0.001)	6.25–30.82
**Fever, N (%)**	77 (25.5 %)	59 (22.01%)	18 (52.94%)	<0.001	3.98 (<0.001)	1.92– 8.29
**Cough, N (%)**	120 (39.74%)	103 (38.43%)	17 (50 %)	0.19		
**Chest pain, N (%)**	12 (3.97%)	9 (3.36%)	3 (8.82%)	0.12		
**Diarrhoea, N (%)**	32 (10.6 %)	27 (10.07%)	5 (14.71%)	0.41		
**Anorexia, N (%)**	14 (4.64%)	10 (3.73%)	4 (11.76%)	0.036	3.44 (0.047)	1.02–11.65
**Chest X-ray findings, N (%)**				<0.001		
**Not performed**	191 (69.45%)	189 (77.78%)	2 (6.25%)			
**Normal**	28 (10.18%)	27 (11.11%)	1 (3.13%)			
**Bilateral pneumonia**	46 (16.73%)	21 (8.64%)	25 (78.13%)		112.5 (<0.001)	24.87–508.84
**Lobar pneumonia**	10 (3.64%)	6 (2.47%)	4 (12.50%)		63 (<0.001)	9.59–413.67
**Azathioprine, N (%)**	3 (0.98%)	2 (0.74%)	1 (2.94%)	0.22		
**Corticosteroids, N (%)**	17 (5.56%)	17 (6.25%)	0	0.13		
**mTOR inhibitors (mTORi), N (%)**	40 (13.07%)	35 (12.87%)	5 (14.71%)	0.76		
**Mycophenolate mofetil (MMF), N (%)**	184 (60.13%)	154 (56.62%)	30 (88.24%)	<0.001	5.74 (<0.001)	1.97–16.76
**Calcineurin inhibitors (CNIs), N (%)**	189 (61.76%)	177 (65.07%)	12 (35.29%)	<0.001	0.29 (<0.001)	0.14–0.62

**Table 5 viruses-17-00273-t005:** Invasive mechanical ventilation (IMV) in liver transplant recipients (LTR) diagnosed with COVID-19.

					Logistic Regression Analysis	
		Mechanical Ventilation				
	Total (N = 306, 1 Lost Case)	No (N = 301)	Yes (N = 5)	*p*-Value	OR (*p*-Value)	95%CI
**Age (years), median (IQR)**	61 (54–68)	61 (53–68)	68 (62–74)	0.1		
**BMI (kg/m^2^), median (IQR)**	26.3 (23–29)	26.3 (22.7–29)	28 (25.5–29.7)	0.21		
**Male sex, N (%)**	208 (67.97%)	203 (67.44%)	5 (100%)	0.12		
**Female sex, N (%)**	98 (32.03%)	98 (32.56%)	0			
**Number of vaccines at diagnosis**	3 (0–3)	3 (0–3)	0 (0–2)	0.2		
**Vaccination N (%)**	211 (69.18%)	209 (69.67%)	2 (40.00%)	0.15		
**Time (months) from vaccination to the onset of symptoms, median (IQR)**	5.87 (3.23–9.03)	5.87 (3.25–9.02)	4.61 (0.1–9.11)	0.56		
**Time (months) from transplantation to the onset of symptoms, median (IQR)**	94.72 (43.31–185.21)	94.46 (43.31–184.49)	134.75 (78.23–191.93)	0.48		
**Kidney failure (Creatinine ≥ 2 mg/dL)**	23 (7.54%)	23 (7.67%)	0	0.52		
**COPD, N (%)**	13 (4.26%)	13 (4.33%)	0	0.63		
**Asthma, N (%)**	6 (1.97%)	6 (2%)	0	0.75		
**HTN, N (%)**	117 (38.36%)	114 (38.00%)	3 (60%)	0.32		
**Cardiovascular events and/or risk factors**	180 (59.21%)	176 (58.86%)	4 (80%)	0.34		
**T2D, N (%)**	124 (40.66%)	119 (39.67%)	5 (100%)	0.006		
**Altered liver enzymes, N (%)**	20 (6.56%)	20 (6.67%)	0	0.55		
**Smokers, N (%)**	50 (16.78%)	49 (16.72%)	1 (20%)	0.94		
**Asthenia, N (%)**	98 (32.56%)	96 (32.43%)	2 (40%)	0.72		
**Dyspnea, N (%)**	45 (14.95%)	41 (13.85%)	4 (80%)	<0.001	24.87 (0.004)	2.71–228.14
**Fever, N (%)**	77 (25.58%)	74 (25.00%)	3 (60%)	0.075		
**Cough, N (%)**	120 (39.87%)	119 (40.20%)	1 (20%)	0.36		
**Chest pain, N (%)**	12 (3.99%)	12 (4.05%)	0	0.65		
**Diarrhoea, N (%)**	32 (10.63%)	31 (10.47%)	1 (20%)	0.49		
**Anorexia, N (%)**	14 (4.65%)	14 (4.73%)	0	0.62		
**Chest X-ray findings, N (%)**				0.008		
**Not performed**	191 (69.71%)	190 (70.63%)	1 (20%)			
**Normal**	28 (10.22%)	28 (10.41%)	0			
**Bilateral pneumonia**	45 (16.42%)	42 (15.61%)	3 (60%)		13.57 (0.025)	1.37–133.71
**Lobar pneumonia**	10 (3.65%)	9 (3.35%)	1 (20%)		21.11 (0.036)	1.22–365.44
**Azathioprine, N (%)**	3 (0.98%)	2 (0.67%)	1 (20%)	<0.001	37.25 (0.006)	2.78–499.16
**Corticosteroids, N (%)**	17 (5.57%)	17 (5.67%)	0	0.58		
**mTORi, N (%)**	40 (13.11%)	40 (13.33%)	0	0.38		
**MMF, N (%)**	183 (60%)	179 (59.67%)	4 (80%)	0.36		
**CNIs, N (%)**	188 (61.64%)	186 (62%)	2 (40%)	0.32		

**Table 6 viruses-17-00273-t006:** Hospital admission in LTRs diagnosed with COVID-19.

					Logistic Regression Analysis	
		Hospital Admission				
	Total (N = 307)	No (N = 235)	Yes (N = 72)	*p*-Value	OR (*p*-Value)	95%CI
**Age (years), median (IQR)**	61 (54–68)	60 (52–67)	64.5 (56–71.5)	0.002	1.04 (*p* = 0.001)	1.02–1.07
**BMI (kg/m^2^), median (IQR)**	26.3 (23–29)	26.1 (22.5–29)	27.32 (24.2–29.7)	0.009	1.08 (*p* = 0.011)	1.02–1.15
**Male sex, N (%)**	209 (68.08%)	156 (66.38%)	53 (73.61%)	0.25		
**Female sex, N (%)**	98 (31.92%)	79 (33.62%)	19 (26.39%)			
**Number of vaccines at diagnosis**	3 (0–3)	3 (1–4)	0 (0–3)	<0.001	0.63 (*p* < 0.001)	0.53–0.75
**Vaccination N (%)**	211 (68.95%)	178 (76.07%)	33 (45.83%)	<0.001	0.23 (*p* < 0.001)	0.15–0.46
**Time (months) from vaccination to the onset of symptoms, median (IQR)**	5.87 (3.23–9.03)	6.26 (3.28–9.18)	3.61 (2.2–6.79)	0.073		
**Time (months) from transplantation to the onset of symptoms, median (IQR)**	94.46 (41.44–185.21)	93.05 (40.85–177.64)	101.98 (43.74–192.44)	0.26		
**Kidney failure (Creatinine ≥ 2 mg/dL)**	23 (7.52%)	12 (5.13%)	11 (15.28%)	0.004	3.33 (*p* = 0.006)	1.4–7.93
**COPD, N (%)**	13 (4.25%)	7 (2.98%)	6 (8.45%)	0.0.45	3 (*p* = 0.055)	0.97–9.26
**Asthma, N (%)**	6 (1.96%)	6 (2.55%)	0	0.17		
**HTN, N (%)**	117 (38.24%)	78 (33.19%)	39 (54.93%)	<0.001	2.45 (*p* = 0.001)	1.43–4.21
**Cardiovascular events and/or risk factors**	181 (59.34%)	127 (54.27%)	54 (76.06%)	0.001	2.67 (*p* = 0.001)	1.46–4. 89
**T2D, N (%)**	125 (40.85%)	85 (36.17%)	40 (56.34%)	0.002	2.28 (*p* = 0.003)	1.33–3.9
**Altered liver enzymes, N (%)**	20 (6.54%)	16 (6.81%)	4 (5.63%)	0.73		
**Smokers, N (%)**	50 (16.72%)	41 (17.75%)	9 (13.24%)	0.14		
**Asthenia, N (%)**	98 (32.45%)	70 (30.43%)	28 (38.89%)	0.18		
**Dyspnea, N (%)**	73 (24.17%)	62 (26.96%)	11 (15.28%)	0.043	24.56 (*p* < 0.001)	10.91–55.25
**Fever, N (%)**	77 (25.50%)	41 (17.83%)	36 (50.00%)	<0.001	4.61 (*p* < 0.001)	2.6–8.17
**Cough, N (%)**	120 (39.74%)	80 (34.78%)	40 (55.56%)	0.002	2.34 (*p* = 0.002)	1.37–4.01
**Chest pain, N (%)**	12 (3.97%)	3 (1.30%)	9 (12.50%)	<0.001	10.81 (*p* < 0.001)	2.84–41.12
**Diarrhoea, N (%)**	32 (10.60%)	15 (6.52%)	17 (23.61%)	<0.001	4.43 (*p* < 0.001)	2.08–9.42
**Anorexia, N (%)**	14 (4.64%)	8 (3.48%)	6 (8.33%)	0.087		
**Chest X-ray findings, N (%)**				<0.001		
**Not performed**	191 (69.45%)	182 (87.92%)	9 (13.24%)			
**Normal**	28 (10.18%)	19 (9.18%)	9 (13.24%)			
**Bilateral pneumonia**	46 (16.73%)	4 (1.93%)	42 (61.76%)		212.33 (*p* < 0.001)	3.39–27.04
**Lobar pneumonia**	10 (3.64%)	2 (0.97%)	8 (11.76%)		80.89 (*p* < 0.001)	14.96–437.44
**Azathioprine, N (%)**	3 (0.98%)	1 (0.43%)	2 (2.78%)	0.077		
**Corticosteroids, N (%)**	17 (5.56%)	14 (5.98%)	3 (4.17%)	0.56		
**mTORi, N (%)**	40 (13.07%)	23 (9.83%)	17 (23.61%)	0.002	2.84 (*p* = 0.003)	1.42–5.67
**MMF, N (%)**	184 (60.13%)	132 (56.41%)	52 (72.22%)	0.017	2.01 (*p* = 0.018)	1.13–3.58
**CNIs, N (%)**	189 (61.76%)	155 (66.24%)	34 (47.22%)	0.004	0.46 (*p* = 0.004)	0.27–0.78

**Table 7 viruses-17-00273-t007:** Intensive care unit (ICU) admission in LTRs diagnosed with COVID-19.

					Logistic Regression Analysis	
	Total (N = 306, 1 Lost Case)	Admission to the ICU				
		No (N = 301)	Yes (N = 5)	*p*-Value	OR (*p*-Value)	95%CI
**Age (years), median (IQR)**	61 (54–68)	61 (53–68)	60 (62–74)	0.1		
**BMI (kg/m^2^), median (IQR)**	26.3 (23–29)	26.3 (23–29)	29 (25.5–34.4)	0.21		
**Male sex, N (%)**	208 (67.97%)	203 (67.44%)	5 (100%)	0.12		
**Female sex, N (%)**	98 (32.03%)	98 (32.56%)	0			
**Number of vaccines at diagnosis**	3 (0–3)	3 (0–3)	0 (0–3)	0.2		
**Vaccination N (%)**	211 (69.18%)	209 (69.67%)	2 (40%)	<0.001		
**Time (months) from vaccination to the onset of symptoms, median (IQR)**	5.87 (3.23–9.03)	5.87 (3.25–9.02)	4.61 (0.10–9.11)	0.56		
**Time (months) from transplantation to the onset of symptoms, median (IQR)**	94.46 (41.44–185.21)	95.28 (45.51–185.31)	46.26 (24.16–137.70)	0.12		
**Kidney failure (Creatinine ≥ 2 mg/dL)**	23 (7.54%)	23 (7.67%)	0	0.52		
**COPD, N (%)**	13 (4.26%)	13 (4.33%)	0	0.63		
**Asthma, N (%)**	6 (1.97%)	6 (2.00%)	0	0.75		
**HTN, N (%)**	117 (38.36%)	114 (38%)	3 (60%)	0.32		
**Cardiovascular events and/or risk factors**	180 (59.21%)	176 (58.86%)	4 (80.00%)	0.34		
**T2D, N (%)**	124 (40.66%)	119 (39.67%)	5 (100%)	0.006		
**Altered liver enzymes, N (%)**	20 (6.54%)	19 (6.44%)	1 (9.09%)	0.73		
**Smokers, N (%)**	50 (16.78%)	49 (16.72%)	1 (20%)	0.94		
**Asthenia, N (%)**	98 (32.56%)	96 (32.43%)	2 (40%)	0.72		
**Dyspnea, N (%)**	45 (14.95%)	41 (13.85%)	4 (80.00%)	<0.001	21.17 (<0.001)	5.47–81.85
**Fever, N (%)**	77 (25.58%)	74 (25.00%)	3 (60.00%)	0.075	4.4 (0.014)	1.35–14.3
**Cough, N (%)**	120 (39.87%)	119 (40.2%)	1 (20%)	0.36		
**Chest pain, N (%)**	12 (3.99%)	12 (4.05%)	0	0.65		
**Diarrhoea, N (%)**	32 (10.63%)	31 (10.47%)	1 (20%)	0.49		
**Anorexia, N (%)**	14 (4.65%)	14 (4.73%)	0	0.62		
**Chest X-ray findings, N (%)**				0.008		
**Not performed**	191 (69.71%)	190 (70.63%)	1 (20%)			
**Normal**	28 (10.22%)	28 (10.41%)	0			
**Bilateral pneumonia**	45 (16.42%)	42 (15.61%)	3 (60%)		28.5 (0.002)	3.34–243.27
**Lobar pneumonia**	10 (3.65%)	9 (3.35%)	1 (20%)		21.11 (0.036)	1.22–365.44
**Azathioprine, N (%)**	3 (0.98%)	2 (0.67%)	1 (20%)	<0.001	13.27 (0.041)	1.12–157.67
**Corticosteroids, N (%)**	17 (5.57%)	17 (5.67%)	0	0.58		
**mTORi, N (%)**	40 (13.11%)	40 (13.33%)	0	0.38		
**MMF, N (%)**	183 (60%)	179 (59.67%)	4 (80%)	0.36		
**CNIs, N (%)**	188 (61.64%)	186 (62%)	2 (40%)	0.32		

**Table 8 viruses-17-00273-t008:** Death from COVID-19.

		Death Due to COVID-19			Univariant Survival Analysis	
	Total (N = 307)	No (N = 295)	Yes (N = 12)	*p*-Value	Hazard Ratio (*p*-Value)	95% CI
**Age (years), median (IQR)**	61 (54.00–68.00)	61(53–68)	68 (61.5–71.5)	0.034		
**Age (years), mean (Standard deviation (SD))**	59.03 (14.03)	58.72 (14.16)	66.50 (7.24)	0.06		
**BMI (kg/m^2^), median (IQR)**	26.30 (23–29)	26.10(22.9–29)	29.05 (26.8–35.25)	0.008		
**BMI (kg/m^2^), mean (SD)**	26.35 (4.68)	26.17 (4.56)	30.54 (5.59)	0.001		
**Obesity, N (%)**	62 (20.67%)	56 (19.44%)	6 (50.00%)	0.01	3.94 (0.002)	1.27–12.22
**Male sex, N (%)**	209 (68.08%)	199 (67.46%)	10 (83.33%)	0.25		
**Doses of vaccination at diagnosis**	3 (0–3)	3 (0–3)	0 (0–1.5)	0.002	0.56 (0.006)	0.32–0.83
**Vaccine (doses)**				0.062		
0	93 (30.39%)	85 (28.91%)	8 (66.67%)			
1	16 (5.23%)	15 (5.1%)	1 (8.33%)			
2	38 (12.42%)	36 (12.24%)	2 (16.67%)			
3	92 (30.07%)	91 (30.95%)	1 (8.33%)			
4	54 (17.65%)	54 (18.37%)	0			
5	13 (4.25%)	13 (4.42%)	0			
**Kidney injury (creatinine ≥ 2 mg/dL)**	23 (7.52%)	19 (6.46%)	4 (33.33%)	<0.001	6.9 (0.002)	2.08–23.02
**Time (months) from transplantation to the onset of symptoms, median (IQR)**	94.46 (41.44–185.21)	94.30 (40.85–185.31)	99.48 (71.66–134.39)	0.82		
**COPD, N (%)**	13 (4.25%)	11 (3.74%)	2 (16.67%)	0.030	4.64 (0.047)	1.02–21.22
**Asthma, N (%)**	6 (1.96%)	6 (2.04%)	0 (0.00%)	0.62		
**HTN, N (%)**	117 (38.24%)	109 (37.07%)	8 (66.67%)	0.039		
**Cardiovascular events and/or risk factors**	181 (59.34%)	170 (58.02%)	11 (91.67%)	0.020		
**T2D, N (%)**	125 (40.85%)	115 (39.12%)	10 (83.33%)	0.002	7.39 (0.01)	1.62–33.76
**Altered liver enzymes, N (%)**	20 (6.54%)	19 (6.46%)	1 (8.33%)	0.80		
**Smokers**	50 (16.72%)	48 (16.72%)	2 (16.67%)	0.14		
**Azathioprine, N (%)**	3 (0.98%)	3 (1.02%)	0	0.73		
**Corticosteroids, N (%)**	17 (5.56%)	16 (5.44%)	1 (8.33%)	0.67		
**mTORi, N (%)**	40 (13.07%)	39 (13.27%)	1 (8.33%)	0.62		
**MMF, N (%)**	184 (60.13%)	172 (58.5%)	12 (100%)	0.004		
**CNIs, N (%)**	189 (61.76%)	187 (63.61%)	2 (16.67%)	0.001	0.12 (0.006)	0.026–0.55
**Period**				<0.001		
1th wave	24 (7.82%)	23 (7.80%)	1 (8.33%)			
2nd wave	40 (13.03%)	36 (12.20%)	4 (33.33%)			
3rd wave	21 (6.84%)	20 (6.78%)	1 (8.33%)			
4th wave	14 (4.56%)	11 (3.73%)	3 (25.00%)			
5th wave	10 (3.26%)	8 (2.71%)	2 (16.67%)			
6th wave	63 (20.52%)	62 (21.02%)	1 (8.33%)			
7th wave	135 (43.97%)	135 (45.76%)	0 (0.00%)			

**Table 9 viruses-17-00273-t009:** Multivariate analysis of respiratory failure and hospital admission.

	Respiratory Failure		Hospital Admission	
	Area Under ROC Curve = 0.851		Area Under ROC Curve = 0.896	
	OR (*p*-Value)	95%CI	OR (*p*-Value)	95%CI
Age	1.056 (0.045)	1–1.1		
Vaccination	0.16 (<0.001)	0.072–0.37	0.2 (<0.001)	0.09–0.44
Kidney injury	5.33 (0.006)	1.62–17.52	4.29 (0.013)	1.35–13.58
HTN	3.69 (0.002)	1.61–8.45	3.25 (0.002)	1.54–6.89
Dyspnea at diagnosis			18.83 (<0.001)	7.61–46.56
Fever at diagnosis			4.08 (<0.001)	1.9–8.74
mTORi			2.8 (0.036)	1.07–7.32
MMF	2.73 (0.008)	1.49–14.71		

**Table 10 viruses-17-00273-t010:** Evolution of severe disease throughout the COVID-19 waves.

									*p*
Period		1st Wave	2nd Wave	3rd Wave	4th Wave	5th Wave	6th Wave	7th Wave	
**Number of cases**		24/307 (7.82%)	40/307 (13.03%)	21/307 (6.84%)	14/307 (4.56%)	10/307 (3.25%)	63/307 (20.52%)	135/307 (43.97%)	
**Respiratory failure**	34/307 (11.07%)	3 (12.50%)	12 (30%)	3 (14.29%)	6 (42.86%)	4 (40%)	4 (6.35%)	2 (1.48%)	<0.001
**IMV**	5/307 (1.63%)	0	2 (5%)	0	1 (7.69%)	0	1 (1.59%)	1 (0.74%)	0.29
**Admission to the ICU**	12/307 (3.91%)	1 (4.17%)	2 (5%)	0	2 (14.29%)	0	5 (7.94%)	2 (1.48%)	0.11
**Hospitalization**	72/307 (23.45%)	11 (45.83%)	16 (40%)	7 (33.33%)	9 (64.29%)	5 (50%)	11 (17.46%)	13 (9.63%)	<0.001
**Death due to COVID-19**	12/307 (3.91%)	1 (4.17%)	4 (10%)	1 (4.76%)	3 (21.43%)	2 (20%)	1 (1.59%)	0 (0.00%)	<0.001
**Severity**									<0.001
Not severe COVID-19	243/307 (79.15%)	15 (62.50%)	24 (60%)	15 (71.43%)	6 (42.86%)	6 (60)	52 (82.54%)	125 (92.59%)	
Severe COVID-19	64/307 (20.85%)	9 (37.50%)	16 (40%)	6 (28.57%)	8 (57.14%)	4 (40%)	11 (17.46%)	10 (7.41%)	

## Data Availability

The original contributions presented in this study are included in the article. Further inquiries can be directed to the corresponding author(s).

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
