# Peer review of "The Impact and Evolution of COVID-19 on Liver Transplant Recipients Throughout the Pandemic “Waves” in a Single Center"

_viruses, 2025, doi:10.3390/v17020273_

Round 1

Reviewer 1 Report

Comments and Suggestions for Authors

The manuscript presents a novel and well-structured study, making a valuable contribution to the understanding of liver transplant recipients during the COVID-19 pandemic.

  • In Figure 2, the y-axis should be adjusted to range from 0.75 to 1.05 to enhance the clarity of the pattern among the data points.
  • The ethical permission number for the utilized data must be included in the Methods section.
  • The discussion should be expanded to establish a stronger correlation between antibiotics used during COVID-19, during/ post liver transplantation, and calcineurin.

Author Response

  • Comments 1: The manuscript presents a novel and well-structured study, making a valuable contribution to the understanding of liver transplant recipients during the COVID-19 pandemic. In Figure 2, the y-axis should be adjusted to range from 0.75 to 1.05 to enhance the clarity of the pattern among the data points.

Response 1: Thank you for your comments. Regarding your first indication, we completely agree, so we changed Figure 2 following these indications.

PAGE 18. Line 642.

  • Comments 2: The ethical permission number for the utilized data must be included in the Methods section.

Response 2: Thank you for pointing this out. The ethical permission reference number has been included at the end of Patients and Methods section in an independent subsection which we have numbered, as it follows:

  1. Ethical and Regulatory Approval. The local Clinical Research Ethics Committee approved the study protocol (ref. no. 20/151) and granted a waiver of informed consent due to its retrospective observational design.

PAGE 5. Lines 246 – 248.

  • Comments 3: The discussion should be expanded to establish a stronger correlation between antibiotics used during COVID-19, during/ post liver transplantation, and calcineurin.

Response 3: Thank you for your comments. We specified that antibiotics were prescribed and adjusted to antibiogram when bacterial superinfection was suspected according to radiological findings and clinical course. To clarify possible doubts, we added to the Patients and Methods section: There was no immunosuppressive adjustment depending on antibiotic administration, but all the patients had repeated immunosuppression level controls to maintain appropriate dosing.

PAGE 11. Lines 582 – 584.

Reviewer 2 Report

Comments and Suggestions for Authors

The article describes the impact of COVID-19 on liver transplant patients in a single center, Hospital 12 de Octubre, in Madrid, Spain. The article's rationale and the patient's medical methodology are adequate. The manuscript is clinically relevant; however, there are some minor issues to consider. The title can be shortened and specify a single center. The second issue is the limitations of the study that should be stated. In addition, the discussion is well-focused, but it may include the patients' years of transplant and immunosuppressive treatment, which may be a factor in preventing the cytokine storm. Another point is gender influence, which may be further analyzed. In general a very good manuscript. 

Author Response

Reviewer 2

  • Comments 1: The article describes the impact of COVID-19 on liver transplant patients in a single center, Hospital 12 de Octubre, in Madrid, Spain. The article's rationale and the patient's medical methodology are adequate. The manuscript is clinically relevant; however, there are some minor issues to consider. The title can be shortened and specify a single center.

Response 1: To shorten the title and to follow this specification we changed the title from The impact of COVID-19 disease on liver transplant recipients. Evolution of infection, prevention and treatment throughout the pandemic "waves, to The impact and evolution of COVID-19 disease on liver transplant recipients throughout the pandemic "waves” in a single center.

PAGE 1. Line 1.

PAGE 3. Line 110.

  • Comments 2: The second issue is the limitations of the study that should be stated.

Response 2: Thank you for pointing this out. We added the study limitations to the manuscript at the end of the discussion, as it follows:

Study limitations. This study has limitations due to its observational and retrospective design. Data were missing as its report in the electronic medical record was not routinely and prospectively collected for research, which led to the exclusion of potential study subjects. Recall bias may have occurred as participants may inaccurately remember their experiences as they were surveyed after the infection, with exception of those admitted to the hospital. Despite collecting data from a considerable number of patients, this study evaluates a cohort in a single center, which limits its external validity and implies a selection bias.

  • Comments 3: In addition, the discussion is well-focused, but it may include the patients' years of transplant and immunosuppressive treatment, which may be a factor in preventing the cytokine storm.

Response 3: As reported in “Results - Section 2. Symptoms”, the median time from transplantation to diagnosis was 94.46 months (IQR 41.44 - 185.21). We had analyzed the relationship between the time from transplantation to the diagnosis and severe COVID-19 scenarios. We show the results in the table in the document attached. The patients’ years of transplant showed no differences between patients that suffered severe COVID-19 disease.

Time from transplantation to diagnosis (months), median (IQR)

A)      Severe COVID-19

Yes

No

p-value

Respiratory failure, , N (%)

34/307 (11.07%)

95.08 (51.28-188.75)

94.46 (41.44-184.49)

0.79

IMV, N (%)

5/307 (1.63%)

134.75 (78.23-191.93)  

94.46 (43.31-184.49)       

0.48

Admission to the ICU, N (%)

12/307 (3.91%)

46.26 (24.16-137.70)

95.28 (45.51-185.31)

0.12

Hospitalization, N (%)

72/307 (23.45%)

101.98 (43.74-192.44)       

93.05 (40.85-177.64)       

0.26

Death due to COVID-19, N (%)

12/307 (3.91%)

99.48 (71.66-134.39)

94.30 (40.85-185.31)

0.82

We modified tables: 4 (PAGE 13), 5 (PAGE 14), 6 (PAGE 15) and 7 (PAGE 16) to clarify this information.

As indicated in PAGE 9. Lines 464 – 466, The median time between receiving the transplant and COVID-19 diagnosis is variable, from 5.7 to 13.1 years. Guarino M et al. observed it was significantly shorter in patients with asymptomatic COVID-19. These, however, did not show any differences in regard to comorbidities or immunosuppressive treatment.

  • Comments 4: Another point is gender influence, which may be further analyzed.

Response 4: Thank you for bringing up this subject.

In the supplementary tables we illustrate the gender influence in the five described severe COVID-19 scenarios. Female sex resulted a protective factor to developing respiratory insufficiency or hospital admission (PAGE 10. Line 500).

All the patients admitted to the ICU and those who required IMV were male as well as 10/12 (83.33%) patients who died from COVID-19.

We have added this data to the discussion (PAGE 10. Lines 527 – 533). This correlates with published literature, being male sex a previously described risk factor for mortality.
